# Small Structural Differences in Proline-Rich Decapeptides Have Specific Effects on Oxidative Stress-Induced Neurotoxicity and L-Arginine Generation by Arginosuccinate Synthase

**DOI:** 10.3390/ph17070931

**Published:** 2024-07-11

**Authors:** Carlos Alberto-Silva, Brenda Rufino da Silva, Julio Cezar Araujo da Silva, Felipe Assumpção da Cunha e Silva, Roberto Tadashi Kodama, Wilmar Dias da Silva, Maricilia Silva Costa, Fernanda Calheta Vieira Portaro

**Affiliations:** 1Natural and Humanities Sciences Center (CCNH), Experimental Morphophysiology Laboratory, Federal University of ABC (UFABC), São Bernardo do Campo 09606-070, SP, Brazil; brendarufino33@gmail.com (B.R.d.S.); julio_cezzar133@hotmail.com (J.C.A.d.S.); fhellcunha@gmail.com (F.A.d.C.e.S.); 2Structure and Functions of Biomolecules Laboratory, Butantan Institute, São Paulo 05503-900, SP, Brazil; pararoberval@gmail.com (R.T.K.); fernanda.portaro@butantan.gov.br (F.C.V.P.); 3Laboratory of Immunochemistry, Butantan Institute, São Paulo 05503-900, SP, Brazil; wilmar.silva@butantan.gov.br; 4Instituto de Pesquisa & Desenvolvimento—IP&D, Universidade do Vale do Paraíba—UNIVAP, Av. Shishima Hifumi, 2911, São José dos Campos 12244-390, SP, Brazil; mscosta@univap.br

**Keywords:** neuroprotective, bioactive peptide, proline-rich peptide, snake venom, oxidative stress, PC12 cells

## Abstract

Introduction. The proline-rich decapeptide 10c (Bj-PRO-10c; ENWPHPQIPP) from the *Bothrops jararaca* snake modulates argininosuccinate synthetase (AsS) activity to stimulate L-arginine metabolite production and neuroprotection in the SH-SY5Y cell line. The relationships between structure, interactions with AsS, and neuroprotection are little known. We evaluated the neuroprotective effects of Bj-PRO-10c and three other PROs (Bn-PRO-10a, <ENWPRPKIPP; Bn-PRO-10a-MK, <ENWPRPKIPPMK; and, Bn-PRO-10c, <ENWPRPKVPP) identified from *Bitis nasicornis* snake venom, with a high degree of similarity to Bj-PRO-10c, on oxidative stress-induced toxicity in neuronal PC12 cells and L-arginine metabolite generation via AsS activity regulation. Methods. Cell integrity, metabolic activity, reactive oxygen species (ROS) production, and arginase activity were examined after 4 h of PRO pre-treatment and 20 h of H_2_O_2_-induced damage. Results. Only Bn-PRO-10a-MK and Bn-PRO-10c restored cell integrity and arginase function under oxidative stress settings, but they did not reduce ROS or cell metabolism. The MK dipeptide in Bn-PRO-10a-MK and valine (V8) in Bn-PRO-10c are important to these effects when compared to Bn-PRO-10a. Bj-PRO-10c is not neuroprotective in PC12 cells, perhaps because of their limited NMDA-type glutamate receptor activity. The PROs interaction analysis on AsS activation can be rated as follows: Bj-PRO-10c > Bn-PRO-10c > Bn-PRO-10a-MK > Bn-PRO-10a. The structure of PROs and their correlations with enzyme activity revealed that histidine (H5) and glutamine (Q7) in Bj-PRO-10c potentiated their affinity for AsS. Conclusions. Our investigation provides the first insights into the structure and molecular interactions of PROs with AsS, which could possibly further their neuropharmacological applications.

## 1. Introduction

Peptides derived from venom have been utilized as a basis to develop potential therapeutic candidates and novel treatments [1,2]. There is growing evidence indicating that peptides derived from natural sources or their synthetic counterparts, including snake venoms, are potential options for neuroprotection [3,4]. Neuroprotective activity of low molecular mass fractions (LMMFs) obtained from snake venoms containing components of the Viperidae family species were reported in different experimental models of neurodegenerative disease [5,6,7,8]. Martins and collaborators found that the LMMF (<14 kDa) of *Bothrops atrox* snake venom was neuroprotective by reducing mitochondrial permeability transition (MPT) and inhibiting neuronal apoptosis [5]. Similarly, the LMMF (<10 kDa) obtained from *Bothrops jararaca* (Bj-LMMF) snake venom showed neuroprotective efficacy against H_2_O_2_-induced oxidative stress in different neuronal cell line type [6,7]. Neuronal PC12 cells [7] and primary cultured hippocampus cells [6] exhibited neuroprotective effects against oxidative stress in response to Bj-LMMF. However, no effects were observed in neuronal SH-SY5Y cells [9] or C6 astrocyte-like cells [7]. These results suggest that the cell line types used to investigate snake venom compound-mediated neuroprotection show important physiological differences with the cell type from which they were derived, and can influence the neuroprotective effects distinct against oxidative stress.

The LMMF of snake venom contains several bioactive peptides with pharmacological relevance [10,11,12]. These include proline-rich oligopeptides (PROs), which are also called bradykinin-potentiating peptides [13,14,15,16]. In general, PROs contain 5 to 14 amino acid residues, with a pyroglutamic residue (<E) at the N-terminal and a proline (P) residue at the C-terminal [14]. PROs longer than seven amino acids share similar features, including a high content of proline (P) residues and the tripeptide sequence Ile–Pro-Pro (IPP) at the C-terminal [14]. Moreover, some PROs show additional dipeptides at the C-terminal of the IPP sequence, such as Met-Lys (MK) reported in *Bitis* sp. snake venom [17] or Ala-Pro (AP) in *Bothrops jararacussu* [18]. Angiotensin I-converting enzyme (ACE; EC 3.4.15.1) inhibition and consequent bradykinin (Bk) potentiation were assumed to be the conventional mechanisms behind the hypotensive effects of numerous PROs [19]. However, new biological activities and targets have been described for PROs, such as argininosuccinate synthetase (AsS; EC 6.3.4.5) activators [20,21,22] and M1 muscarinic acetylcholine receptor (M1 mAChR) agonists [23,24,25].

PRO-mediated neuroprotection with different structural and functional features was reported on regarding oxidative stress-conditioned toxicity in the human neuroblastoma SH-SY5Y cell line [9,26] and neuronal PC12 cells [25,27]. Remarkably, the specific impacts of PROs on toxicity caused by oxidative stress have indicated that they can influence their targets through a range of distinct approaches [4]. Bj-PRO-7a (<EDGPIPP) and Bj-PRO-10c (<ENWPHPQIPP) enhanced the survival of SH-SY5Y cells and reduced the production of reactive oxygen species (ROS), lipid peroxidation, and total glutathione in response to H_2_O_2_-induced damage [9,26]. Nevertheless, the molecular pathways that explain the neuroprotection induced by both peptides are independent. The neuroprotective effect of Bj-PRO-7a on PC12 cells seems to be mediated via the activation of mAChR-M1 through the G-protein/phospholipase C (PLC)/protein kinase C (PKC) signaling pathway. This activation leads to the inhibition of glycogen synthase kinase 3 beta (GSK3β), resulting in a decrease in oxidative stress and neuronal damage [25].

Conversely, the neuroprotective properties of Bj-PRO-10c have been related to the enhancement of AsS expression and activity, which in turn improves the synthesis of L-arginine [4,9]. AsS catalyzes argininosuccinate formation through aspartate and citrulline conjugation [4,28]. Argininosuccinate lyase (AsL; EC 4.3.2.1) cleaves argininosuccinate to produce fumarate and L-arginine [28]. L-arginine can be catabolized by four sets of enzymes: nitric oxide synthases (NOS; EC 1.14.13.39), arginases (EC 3.5.3.1), arginine:glycine amidinotransferase (EC 2.1.4.1), and arginine decarboxylase (ADC; EC 4.1.1.19) [29]. Products of L-arginine metabolism were reported in important neuroprotection pathways [29,30,31]. Agmatine is a compound synthesized by ADC acting on the N-methyl-D-aspartate (NMDA)-type glutamate receptor, α2-adrenoceptors, and imidazoline receptors, elucidating its role in enhancing cell viability, neuronal protection, and synaptic plasticity [32]. The L-arginine hydrolytic cleavage by arginase produces urea and ornithine, which follow to produce polyamines (putrescine, spermine, and spermidine) implicated in neuroprotection pathways [33]. For these reasons, it has been hypothesized that Bj-PRO-10c enhances L-arginine synthesis by activating AsS and that agmatine or polyamine generation explains its neuroprotective action against oxidative stress in neuronal SH-SY5Y cells [9]. The dependence of this function on its basic structure is well established [9], however the specific amino acids crucial for maintaining its activity on AsS remain unknown.

The purpose of this study was to investigate if Bj-PRO-10c (<ENWPHPQIPP) and the other three PROs with a high degree of similarity to Bj-PRO-10c (Bn-PRO-10a, <ENWPRPKIPP; Bn-PRO-10a-MK: <ENWPRPKIPPMK; and Bn-PRO-10c, <ENWPRPKVPP) identified from *Bitis nasicornis* snake venom [17] were also able to protect neuronal PC12 cells against oxidative stress-induced alterations. PC12 cells are derived from a pheochromocytoma of the adrenal medulla, which represents typical neuronal cells of the peripheral nervous system with dopaminergic properties and has been reported to be an appropriate model to explore the neuroprotective effects of the snake venom compound [5,7,25]. Furthermore, the involvement of L-arginine metabolite generation via AsS activity regulation produces agmatine or polyamines with neuroprotective properties, considering the small structural differences between the four PROs.

## 2. Results

### 2.1. Toxicological Profile of PROs

Initially, the toxicological impacts of PROs on PC12 cells were examined. Peptides that exhibited minimal or no toxicity were subjected to neuroprotective experiments, as seen in Figure 1. Bn-PRO-10a-MK and Bn-PRO-10c increased cell viability by 109.9 ± 2.23% and 108.3 ± 1.61% after 24 h of treatment with 0.01 μmol·L^−1^, respectively. However, no alterations were observed at 10 and 1 μmol·L^−1^. Both Bn-PRO-10a and Bj-PRO-10c did not exhibit any significant cytotoxic effects (*p* > 0.05) at all tested concentrations. Acrylamide (100 mM) reduced cell integrity in PC12 by 58.77 ± 11.14% (Figure 2).

### 2.2. Neuroprotection against Oxidative Stress

PC12 cells exposed to H_2_O_2_ at 0.5 μmol·L^−1^ for 20 h significantly decreased cell integrity to 74.58 ± 2.16% after treatment, compared to the control (Figure 3). Bn-PRO-10a-MK at doses ranging from 12.5 to 0.19 μmol·L^−1^ had higher cell integrity than the H_2_O_2_-treated group (Figure 3B). Bn-PRO-10c at 12.5 and 6.25 μmol·L^−1^ significantly increased cell viability in relation to cells treated only with H_2_O_2_ (Figure 3C). On the other hand, Bn-PRO-10a and Bj-PRO-10c did show neuroprotective effects against H_2_O_2_-induced oxidative stress (Figure 3A and Figure 3D, respectively). Bn-PRO-10a-MK and Bn-PRO-10c demonstrated neuroprotective action against H_2_O_2_-induced stress in different concentrations, but the concentration of 6.25 μmol·L^−1^ showed the highest statistical significance and was used in the metabolic activity, cell integrity, ROS generation, and arginase activity studies. In addition, the neuroprotection index of Bn-PRO-10a-MK and Bn-PRO-10c at 6.25 μmol·L^−1^ (0.90 and 0.85, respectively) were higher than Bn-PRO-10a and Bj-PRO-10c (0.65 and 0.70, respectively) (Table 1). The H_2_O_2_ effects on cells showed a neuroprotection index of 0.60.

#### 2.2.1. Metabolic Activity

Treatment with Bn-PRO-10a-MK and Bn-PRO-10c had no effect on dehydrogenase-dependent resazurin metabolism (Figure 4B). However, when PC12 cells were exposed to H_2_O_2_, their metabolic activity decreased. Cells treated with Bn-PRO-10a-MK and Bn-PRO-10c showed no change in their response to H_2_O_2_-induced stress compared to the H_2_O_2_ group (Figure 4B).

#### 2.2.2. Cell Integrity

H_2_O_2_-induced neurotoxicity impaired cell integrity, but this was attenuated when cells were treated with Bn-PRO-10a-MK or Bn-PRO-10c (Figure 4C). The cell integrity was maintained after treatment only using Bn-PRO-10a-MK and Bn-PRO-10c in relation to the control.

#### 2.2.3. ROS Generation

ROS levels were considerably greater in the H_2_O_2_ group than in the control group (Figure 4D), whereas the Bn-PRO-10a-MK and Bn-PRO-10c treatments had ROS levels similar to the control group. Surprisingly, Bn-PRO-10a-MK and Bn-PRO-10c did not reduce ROS levels against H_2_O_2_-induced oxidative stress.

#### 2.2.4. Arginase Activity

The oxidative stress caused by H_2_O_2_ treatment reduced arginase activity by reducing urea levels in comparison to the control (Figure 4E). However, compared to the H_2_O_2_ group, Bn-PRO-10a-MK and Bn-PRO-10c increased arginase activity against oxidative stress. Also, both peptides exhibited the ability to enhance arginase activity even in the absence of oxidative stress, which is rather interesting. More specifically, Bn-PRO-10a-MK exhibited a 130.4 ± 4.2% rise in activity, which was significantly lower than the activity of Bn-PRO-10c, which showed 154.1 ± 3.0% of activity.

### 2.3. PROs Do Not Affect the Mitochondrial Membrane Potential (ΔΨm)

Nitroprusside sodium (NPS) significantly reduced ΔΨm in PC12 cells when compared to the control (Figure 5B). However, no alterations were demonstrated when cells were treated with all PROs tested at 6.25 μmol·L^−1^.

### 2.4. PROs Regulate L-Arginine Production and Arginase Activity in a Structure-Dependent Manner

The cells that were exposed to various PROs at a concentration of 6.25 μmol·L^−1^ exhibited increased arginase activity in comparison to the control group, as seen in Figure 5C. Furthermore, Bn-PRO-10c and Bj-PRO-10c had significantly higher activity than Bn-PRO-10a and Bn-PRO-10a-MK. There was no noticeable difference in arginase activity when cells were exposed to (inv)Bj-PRO-10c, which consists of the inverted amino acid sequence of Bj-PRO-10c (Figure 5C). The use of MDLA, either alone or in combination with PROs, resulted in a decrease in arginase activity compared to the control group (Figure 5D). However, arginase activity exhibited considerably lower levels in cells treated with MDLA and Bn-PRO-10c or Bj-PRO-10c compared to Bn-PRO-10a or Bn-PRO-10a-MK. In addition, MDLA, either alone or in combination with Bn-PRO-10a, Bn-PRO-10a-MK, and Bn-PRO-10c, did not alter the integrity of the cells. Unlike all of this, Bj-PRO-10c enhanced cell integrity compared to other PROs and the control group (Figure 5E).

## 3. Discussion

Bj-PRO-10c, a bioactive proline-rich decapeptide isolated from *Bothrops jararaca* snake venom [13], has been widely described as an AsS activator, increasing the availability of L-arginine in different cell types [19,20,21,22]. PROs with different primary structures, including Bj-PRO-5a (<EKWAP), Bj-PRO-9a (<EWPRPQIPP), Bj-PRO-11e (<EARPPHPPIPP), Bj-PRO-12b (<EWGRPPGPPIPP), and Bj-PRO-13a (<EGGWPRPGPEIPP), were also assessed for the study of their impact on AsS activity [21]. The influence of these PROs on AsS activation showed that some peptides are efficient activators, whereas others are not. They may be ranked in the following order: Bj-PRO-10c > Bj-PRO-13a > Bj-PRO-12b > Bj-PRO-5a > Bj-PRO-9a/Bj-PRO-11e. It is suggested that the fundamental structure of Bj-PRO-10c plays a crucial role in its effects on AsS activity [21]. Here, the major finding of this study was that small variations in the amino acid sequences of Bj-PRO-10c when compared with Bn-PRO-10a, Bn-PRO-10a-MK, and Bn-PRO-10c, the bioactive oligopeptides described from *Bitis nasicornis* crude venom [17], exhibit distinct effects on the AsS and arginase activities in neuronal PC12 cells. The structure analysis and biological activities of Bj-PRO-10c compared to Bn-PRO-10a, Bn-PRO-10a-MK, and Bn-PRO-10c, suggested that histidine—position 5 (H5) and glutamine—position 7 (Q7) are important amino acid residues to increase its interaction affinity using AsS (Table 1). We also demonstrated that these differences reflect on their neuroprotective properties in PC12 cells using an H_2_O_2_-induced oxidative stress model for the study of neurodegenerative diseases in vitro.

Oxidative stress caused by H_2_O_2_ has been used as an in vitro model for neurodegeneration which leads to mitochondrial failure, lipid peroxidation, alterations in the cell membrane, and cell death [34,35]. H_2_O_2_ causes a decrease in cell viability in a way that is dependent on its concentration when applied to neuronal PC12 cells [36,37,38,39], and it is commonly employed to study the protective effects of venom compounds from various species against oxidative stress [7,36,39]. Neuroprotection provided by Bj-PRO-7a against H_2_O_2_-induced damage was shown in SH-SY5Y cells obtained from central nervous system (CNS) tissue [40,41], and PC12 cells produced from peripheral nervous system (PNS) tissue [25]. Bj-PRO-10c-mediated neuroprotection was shown in SH-SY5Y cells [9], but curiously this effect was not detected in PC12 cells. However, despite the significant similarities among the PROs tested, only Bn-PRO-10a-MK and Bn-PRO-10c exhibited a dose-dependent capacity to restore cell viability against oxidative stress damage, in contrast to Bn-PRO-10a and Bj-PRO-10c.

Bj-PRO-10c-mediated neuroprotection in SH-SY5Y cells is attributed to enhanced L-arginine production, which is accomplished via upregulating AsS [9]. Elevated L-arginine levels can stimulate the production of agmatine or other polyamines [42], which was demonstrated in numerous CNS cell lines and animal models to be neuroprotective against different types of stress such as excitotoxicity and oxidative damage [32,33,43,44]. Studies indicate that agmatine and other polyamines can inhibit glutamate receptors, offering protection from neurotoxic stresses through interactions with the NMDA receptor [30,32,33,43]. Thus, it has been proposed that the neuroprotection mechanisms of Bj-PRO-10c are related to the activation of the L-arginine metabolite pathway, and their products could reduce NMDA receptor activity and stress oxidative markers [4,9]. PC12 cells produce mRNA for NMDA receptor subunits, but only a small amount of receptor protein subunits are present, and there are no functional NMDA-operated channels in this cell line [45,46]. Our results demonstrated that Bj-PRO-10c increases L-arginine production by AsS and arginase activity in PC12 cells, and possibly increases agmatine and other polyamines by ornithine metabolism. However, because these cells do not have functional NMDA receptors and Bj-PRO-10c has higher levels of polyamines, its neuroprotective effects against oxidative stress could be ineffective explaining why this peptide could fail to provide neuroprotection in PC12 cells. Curiously, Bn-PRO-10c (<ENWPRPKVPP) and Bj-PRO-10c (<ENWPHPQIPP) show different biological effects (Table 1), suggesting that, aside from both peptides increasing L-arginine production, Bn-PRO-10c-mediated neuroprotection appears to be independent of the L-arginine metabolism pathway reported by Bj-PRO-10c. Further research will be needed to gain a more comprehensive understanding of the correlation between the structure and functions of PROs, as well as their ability to protect against oxidative stress.

The toxicity of compounds is a critical part of current pharmaceutical research [47]. In our study, we assessed the effects of PROs on neuronal PC12 cell integrity and found no cytotoxic effects in the tested experimental conditions. However, Bn-PRO-10a-MK and Bn-PRO-10c at 0.01 μmol·L^−1^ increased cell viability after 24 h of treatment. Also, both peptides only restored cell integrity against H_2_O_2_-induced toxicity in a concentration-dependent manner, and a concentration of 6.25 μmol·L^−1^ displayed the highest statistical significance and was used to explore their effects on other oxidative stress markers. Neuroprotection mediated by venom snake-derived peptides such as p-BTX-I, FPWIIS-NH_2_, Bj-PRO-7a, and Bj-PRO-10c significantly decreased ROS generation in response to oxidative stress conditions in several cell types [4,8,9,25,26,27]. In the current investigation, Bn-PRO-10a-MK and Bn-PRO-10c did not reduce ROS production or cell metabolism, but restored cell integrity and arginase activity against oxidative stress-induced alterations in PC12 cells. Neuroprotective peptides should be able to prevent neuronal loss while also maintaining structure and function, hence decreasing cell damage and death [3]. Based on the oxidative stress markers tested, Bn-PRO-10a-MK and Bn-PRO-10c cannot be considered neuroprotective in PC12 cells and will need to be studied in more detail. Nevertheless, both peptides increased arginase activity without oxidative stress conditions, including statistical differences between them. These findings encouraged further examination of the other two peptides, Bj-PRO-10c and Bn-PRO-10a, on AsS and arginase activity.

L-Arginine is synthesized from citrulline by the sequential action of the AsS and AsL enzymes [29]. AsS inhibition reduces L-arginine bioavailability and, consequently, the generation of its metabolites [28], as demonstrated in our work when PC12 cells were treated with MDLA or as reported in the literature [7,25]. Many reports have demonstrated that Bj-PRO-10c interacts with AsS, promoting its activation and increasing the production of L-arginine and its metabolites [20,21,22,48,49,50]. When we examined how the various PROs impacted arginase activity and cell integrity in PC12 cells treated or untreated with MDLA, we found that Bj-PRO-10c interacted with AsS more effectively than the others. This makes more L-arginine, which is then turned into urea, which was used as a marker to measure arginase activity in this study. Remarkably, only Bj-PRO-10c showed an improvement in the number of cells when AsS was inhibited by MDLA, equivalent to what was seen in Bj-LMMF [7]. Despite that, neither of the PROs altered the mitochondrial membrane potential, indicating that these peptides mainly influence AsS activity.

Bj-PRO-10c is internalized by different cell lines [20,22,51] and appears to interact directly with AsS [22]. Studies about the contribution of individual proline to AsS activation by Bj-BPP-10c were evaluated by the substitution of each proline residue located in the peptide sequence by Ala residues [20]. Guerreiro and colleagues (2009) demonstrated that all individual substitutions had a partial impact on Bj-BPP-10c’s ability to enhance AsS enzymatic activity, whereas the C-terminal proline residues appeared to be essential for peptide engagement with AsS [20]. All PROs studied in the present study have the <ENWP sequence at the N-terminal, and the positions of proline (P) residues are also conserved (Table 1). Despite the high similarity of PROs, the small differences, particularly in the middle region of the peptide or near the C-terminal of Bn-PRO-10a-MK, are responsible for their distinct impacts on oxidative stress-induced toxicity in neuronal PC12 cells and the production of L-arginine by AsS (Table 1). The presence of MK dipeptide in the C-terminal of Bn-PRO-10a-MK restored oxidative stress-induced neurotoxicity, but not in Bn-PRO-10a. In addition, the presence of V8 in Bn-PRO-10c partially reversed the toxicity, but Bn-PRO-10a with I8 at the same location did not. Bj-PRO-10c has H5 and Q7, whereas other Bn-PROs have R7 and K7 in the same positions. Thus, for the first time, we report that the amino acids H5 and Q7 positioned in the Bj-PRO-10c structure play a crucial role in explaining the increased AsS activity in PC12 cells in relation to the other PROs.

## 4. Materials and Methods

### 4.1. Reagents and Synthetic Peptide

All reagents and chemicals used in the present study were of an analytical reagent grade (purity higher than 95%) and purchased from Calbiochem-Novabiochem Corporation (San Diego, CA, USA), Gibco BRL (New York, NY, USA), Fluka Chemical Corp. (Buchs, Switzerland), or Sigma-Aldrich Corporation (St. Louis, MO, USA). The synthetic peptides [Bn-PRO-10a (<ENWPRPKIPP); Bn-PRO-10a-MK (<ENWPRPKIPPMK); Bn-PRO-10c (<ENWPRPKVPP); Bj-PRO-10c (<ENWPHPQIPP); and the inverted sequence of Bj-PRO-10c (PPIQPHPWNE), named (inv)Bj-PRO-10c, were purchased from FastBio (Ribeirão Preto, Brazil) or GenOne Biotechnologies (Rio de Janeiro, Brazil). The purity of all peptides was analyzed by reversed-phase high-performance liquid chromatography (HPLC; Shimadzu, Kyoto, Japan) and MALDI-TOF mass spectrometry (Amersham Biosciences, Uppsala, Sweden).

### 4.2. Cell Line, Culture and Maintenance

Neuronal PC12 cells derived from a transplantable rat pheochromocytoma (ATCC^®^ CRL-1721™ from the American Type Culture Collection—ATCC, Manassas, VA, USA) were used in the present study. PC12 cells were routinely cultured in a D10 medium [DMEM medium (Sigma-Aldrich, St. Louis, MO, USA)], and supplemented with 10% fetal bovine serum (FBS) (Gibco, Waltham, MA, USA), 1% (*v*·*v*^−1^) of 10,000 U·mL^−1^ penicillin, 10 mg·mL^−1^ streptomycin, and 25 µg·mL^−1^ amphotericin B solutions (Sigma-Aldrich, St. Louis, MO, USA). The cultures were kept at 37 °C in a humidified atmosphere containing 5% CO_2_ and 95% air (Water Jacketed CO_2_ Incubator, Thermo Scientific, Waltham, MA USA). The culture medium was replaced every 2–3 days, and at 80% confluence, cells were passaged using trypsin–EDTA solution [0.05% (*m*·*v*^−1^) trypsin and 0.02% (*m*·*v*^−1^) EDTA]. The cells were cultured and used for the experiments up to a maximum passage number of 30.

### 4.3. Cytotoxicity Studies

Cells were previously seeded into 96-well plates (Nest Biotechnology, Rahway, NJ, USA), at 5 × 10^3^ cells per well and maintained for 24 h in a humidified atmosphere containing 5% CO_2_ and 95% air. The cells were treated with 10, 1, and 0.01 μmol·L^−1^ of PROs in a final volume of 0.10 mL and incubated at 37 °C for 24 h. Untreated cells or cells treated with acrylamide (100 mmol·L^−1^; positive control) diluted in the medium culture were included in all experiments. The cytotoxic effects of PROs were determined by the staining of attached cells with crystal violet dye, according to the literature [52]. After the treatment, the medium was aspirated, and the cells were stained with a crystal violet staining solution (0.5%; *v*·*v*^−1^), washed, and air-dried. Then, methanol (200 μL) was added to each well, and the absorbance was measured at 570 nm using a BioTek Epoch microplate spectrophotometer (BioTek Epoch, Santa Clara, CA, USA). Data were obtained from three independent experiments in triplicate, expressed as the mean ± SD, and represent the percentage of cell viability in relation to the control.

### 4.4. Neuroprotection Assay in PC12 Cells against Oxidative Stress

PC12 cells were seeded at 5 × 10^3^ cells per well in a 96-well plate (Nest Biotechnology, Rahway, NJ, USA) for 24 h. Then, cells were pre-treated for 4 h at 37 °C with different PROs (25 to 0.097 µM), diluted in a D10 medium. After, the mediums were replaced by a medium containing the PROs and H_2_O_2_ (0.5 mmol·L^−1^) oxidative stress, according to descriptions in the literature [7]. The plate was incubated for 20 h more (PROs + H_2_O_2_ group). Cells untreated (control) or treated with H_2_O_2_ were also incubated under the same conditions (Figure 1). Next, the neuroprotective effects against H_2_O_2_-induced oxidative stress of PROs on PC12 integrity cells were estimated using crystal violet dye—described above [52]. Data were obtained from three independent experiments in sextuplicate and expressed as box-and-whisker plots. Additionally, the quantification of the effect of peptides on oxidative stress was conducted using a neuroprotection index (NI). This index assesses the correlation between the viability of cells that were not treated and the viability of cells that were treated with H_2_O_2_ in combination with the peptides. The PROs that rescued the H_2_O_2_-induced neurotoxicity at 6.25 μmol·L^−1^ in the cell integrity assays that displayed the highest statistical significance when compared to the other concentrations evaluated were chosen to investigate PRO-mediated protection on metabolic activity, ROS generation, and arginase activity using concentrations.

#### 4.4.1. Metabolic Activity and Cell Integrity

Resazurin dye (7-hydroxy-3H-phenoxazin-3-one 10-oxide; Sigma-Aldrich, St. Louis, MO, USA) was used as an indicator of metabolic activity by resazurin reduction into resorufin [37]. PC12 cells were exposed to different treatments in the presence of 40 μmol·L^−1^ resazurin diluted in culture medium. After the treatment, resorufin fluorescence was assessed by 530 nm excitation and 590 nm emission in a BioTek Synergy microplate reader (BioTek Synergy HT Multi Mode Microplate Reader, Santa Clara, CA, USA). Wells containing the same volume of solution but no cells were used as blank controls. Data were expressed as box-and-whisker plots of cell metabolism percentages in relation to the control. We also estimated PC12 integrity cells using crystal violet dye, as previously described.

#### 4.4.2. ROS Quantification

ROS generated were assessed using 2′,7′-dichlorodihydrofluorescein diacetate (H_2_DCF-DA; Sigma-Aldrich, St. Louis, MO, USA) staining, according to the previous procedure [53]. An H_2_DCF-DA stock solution was dissolved into anhydrous DMSO before incubation, which was diluted to 1 mM and stored as aliquots in a −20 °C freezer. The stock solution and aliquots were made in the dark. After the treatments, the culture medium was collected and centrifuged at 99,391× *g* for 5 min. Fifty microliters of culture medium were separated and diluted three-fold into a PBS solution supplemented with H_2_DCF-DA (25 μmol·L^−1^) in a 96-well dark plate (SPL Life Science—Gyeonggi-do, Korea). Samples were incubated for 1 h at 37 °C. H_2_DCF-DA fluorescence intensity was measured using a BioTek Synergy microplate reader (BioTek Synergy HT Multi Mode Microplate Reader, Santa Clara, CA, USA). The excitation filter was set at 480 nm and the emission filter at 530 nm. The results of each experiment were reported as mean values from triplicate wells as arbitrary units. Data were expressed as box-and-whisker plots of fluorescence percentages in relation to the control.

#### 4.4.3. Arginase Activity

The arginase activity was determined by measuring the metabolite urea, a byproduct of L-arginine degradation from cells, according to the literature [54]. The medium culture of cells untreated and treated was collected and used to determine the urea concentrations using a urea analysis kit provided by Roche (Roche/Hitachi cobas c systems; Roche Diagnostics Corporation, Indianapolis, IN, USA) and a BioTek Epoch microplate spectrophotometer (BioTek Epoch, Santa Clara, CA, USA) at 340 nm. A calibration curve was prepared with increasing amounts of urea between 20 to 0.04 mmol·L^−1^. Data were expressed as box-and-whisker plots of arginase activity percentages in relation to the control.

### 4.5. Effects of Distinct PROs on Mitochondrial Membrane Potential

PC12 cells were seeded on 96-well plates (Nest Biotechnology, Rahway, NJ, USA) at 5 × 10^3^ per well, and treated with PROs at 6.25 μmol·L^−1^ diluted in a D10 medium (100 µL) for 24 h. After, the medium was removed, each well was washed three times with PBS (pH 7.4), and then 1 μmol·L^−1^ of Tetramethylrhodamine methyl (TMRM— Thermo Scientific, Waltham, MA, USA) was added. TMRM is a potentiometric cell-permeable fluorescent dye that accumulates in the negatively charged interior of mitochondria and has been used to evaluate mitochondrial membrane potential (ΔΨm) [55]. After incubation at 37 °C for 40 min, the cells were washed three times with PBS. The fluorescence was measured in the BioTek Synergy microplate reader (BioTek Synergy HT Multi Mode Microplate Reader, Santa Clara, CA, USA) at an excitation wavelength of 540 nm and emission at 570 nm. Data were obtained from three independent experiments in sextuplicate and expressed as box-and-whisker plots of fluorescence percentages in relation to the control group.

### 4.6. Effects of Similar PROs on Arginase Activity and L-Arginine Production

The effects of Bn-PRO-10a, Bn-PRO-10a-MK, Bn-PRO-10c, Bj-PRO-10c, and (inv)Bj-PRO-10c were also evaluated on arginase indirect activity. PC12 cells seeded in 96-well plates (Nest Biotechnology, Rahway, NJ, USA) at 5 × 10^3^ per well were treated with distinct PROs at 6.25 μmol·L^−1^ diluted in a D10 medium (100 µL) for 24 h. After, the medium culture was used to determine the urea concentrations using a urea analysis kit provided by Roche as previously described. Furthermore, the impact of PROs on L-arginine metabolism was investigated by utilizing a selective inhibitor α-Methyl-DL-aspartic acid (MDLA; Sigma-Aldrich, St. Louis, MO, USA) of argininosuccinate synthase (AsS), a crucial enzyme involved in the conversion of L-citrulline to L-arginine [9,22]. In these experiments, PC12 cells seeded in 96-well plates (Nest Biotechnology, Rahway, NJ, USA) at 5 × 10^3^ per well were pre-treated with MDLA (1 mmol·L^−1^) diluted in D10 (100 µL) for 1 h. After, the solution was replaced by a D10 medium containing PROs (6.25 μmol·L^−1^) for 24 h at 37 °C. We also incubated cells under the same conditions, either untreated (control group) or treated with MDLA. Afterward, all groups were analyzed for arginase activity and cell integrity. Data were obtained from three independent experiments in sextuplicate and expressed as box-and-whisker plots of arginase activity or cell integrity percentages in relation to the control.

### 4.7. Statistical Analyses

Data were shown as the mean ± SD or box-and-whisker plots of three independent experiments in sextuplicate. Data were analyzed using a one-way analysis of variance (ANOVA) for between-group comparisons, followed by a Dunnett’s post-hoc test for multiple comparisons or to compare a number of treatments with a single control. Values of *p* < 0.05 were considered to be statistically significant. The analyses were performed using GraphPad Prism 6.0 software (GraphPad Software, Inc., La Jolla, CA, USA).

## 5. Conclusions

Taken together, we demonstrated that small variations in the amino acid sequences of Bn-PRO-10a, Bn-PRO-10a-MK, Bn-PRO-10c, and Bj-PRO-10c exhibit distinct effects on cell viability against H_2_O_2_-induced oxidative stress in the AsS enzymatic activity in neuronal PC12 cells. Only Bn-PRO-10a-MK and Bn-PRO-10c restored cell integrity and arginase activity in oxidative stress conditions, but they did not reduce ROS production or cell metabolism. It suggests that the presence of the MK dipeptide in Bn-PRO-10a-MK and V8 in Bn-PRO-10c is important to induce these effects when compared to Bn-PRO-10a effects. Additionally, the structure analysis and AsS enzymatic activity of Bj-PRO-10c compared to Bn-PRO-10a, Bn-PRO-10a-MK, and Bn-PRO-10c suggested that H5 and Q7 are important amino acid residues to increase its interactions affinity by AsS. These findings provide relevant information on the specific structure of PROs and their molecular interactions with AsS a crucial enzyme involved in the synthesis of L-arginine and its metabolites for different neuropharmacological applications.

## Figures and Tables

**Figure 1 pharmaceuticals-17-00931-f001:**
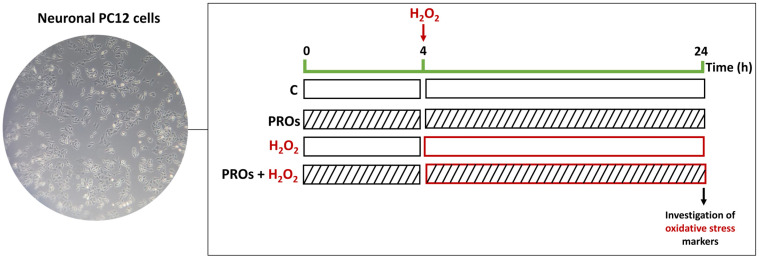
Schematic representation of experimental groups and treatment. The current study employed neuronal PC12 cells, as shown in a typical photomicrograph (magnification: ×80).Cells were pre-treated with a D10 medium or PROs diluted in medium for 4 h at 37 °C. After that, the mediums were replaced by a medium containing PROs or/and H_2_O_2_ (0.5 mM) and incubated for 20 h more. C: control; PROs: proline-rich oligopeptides.

**Figure 2 pharmaceuticals-17-00931-f002:**
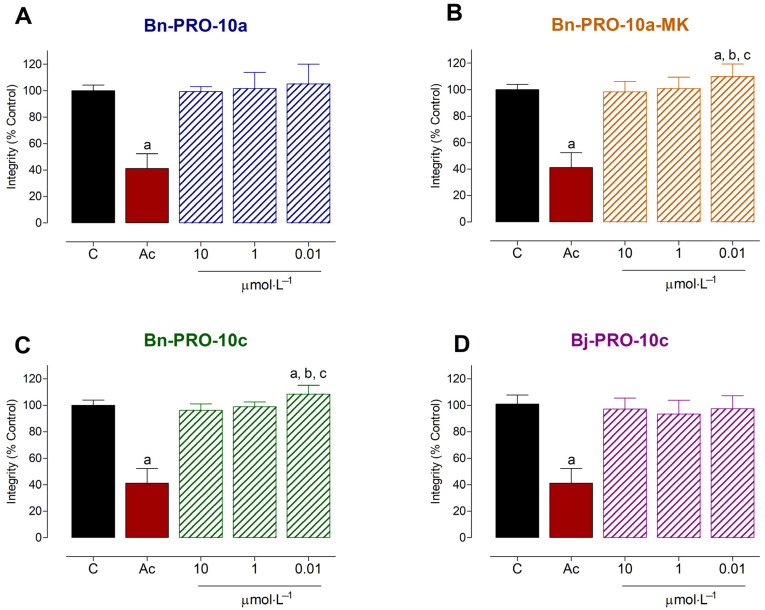
Toxicological profiles of PROs on the cell integrity of neuronal PC12 cells. Cells were treated with (**A**) Bn-PRO-10a, (**B**) Bn-PRO-10a-MK, (**C**) Bn-PRO-10c, and (**D**) Bj-PRO-10c at 10, 1, and 0.01 μmol·L^−1^ for 24 h. Cells without treatment (negative control) and treated with acrylamide (Ac) at 100 mmol·L^−1^ (positive control) were included in all experiments. Values are expressed as the median ± SD from three independent experiments in triplicate and analyzed by a one-way ANOVA followed by Dunnett’s post-test. Statistical differences (*p* < 0.05) were identified by distinct letters: (a) in relation to the control; (b) in relation to 10 μmol·L^−1^; and (c) in relation to 1 μmol·L^−1^.

**Figure 3 pharmaceuticals-17-00931-f003:**
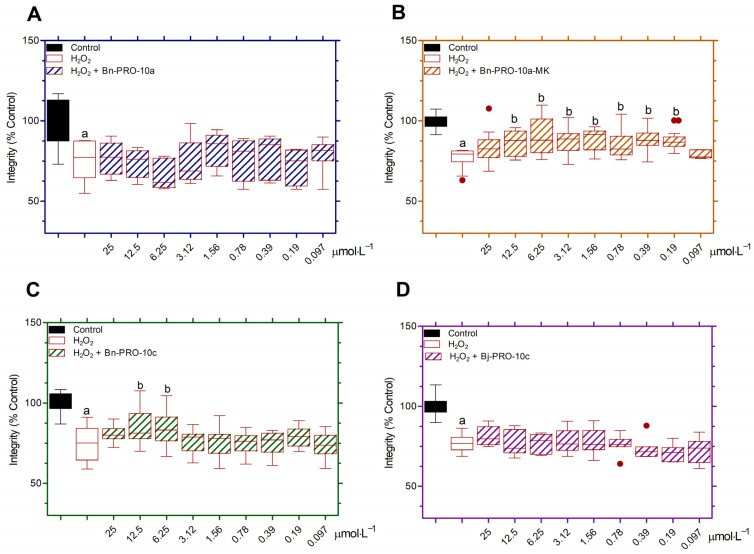
PROs-mediated neuroprotection on oxidative stress-induced changes in neuronal cells. The protective effects of (**A**) Bn-PRO-10a, (**B**) Bn-PRO-10a-MK, (**C**) Bn-PRO-10c, and (**D**) Bj-PRO-10c were assessed at different concentrations against oxidative stress-induced neurotoxicity on the integrity of PC12 cells, using crystal violet dye. Values were presented as box-and-whisker plots from three independent experiments in sextuplicate. Data were analyzed by a one-way ANOVA followed by Dunnett’s post-test. Statistical differences (*p* < 0.05) were identified by distinct letters: (a) in relation to the control; and (b) in relation to H_2_O_2_.

**Figure 4 pharmaceuticals-17-00931-f004:**
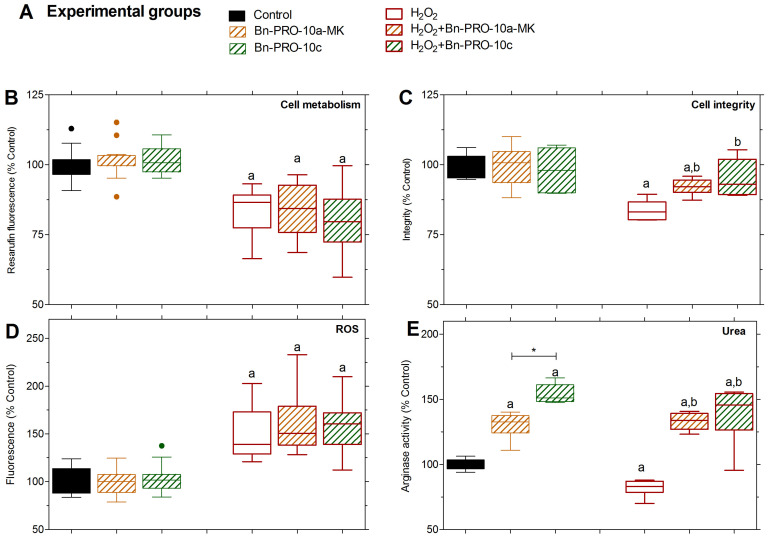
The neuroprotective effects of Bn-PRO-10c and Bn-PRO-10a-MK on oxidative stress markers. (**A**) Schematic representation of experimental groups tested. (**B**) Resazurin reduction into resorufin was used to estimate the metabolic activity of cells. (**C**) Effects on cell integrity were measured by crystal violet dye. (**D**) Reactive oxygen species (ROS) generation induced by H_2_O_2_ and treatments were performed using the 2′,7′-dichlorodihydrofluorescein diacetate (H_2_DCF-DA) assay. (**E**) Arginase activity was determined by measuring the metabolite urea, a byproduct of L-arginine degradation from PC12 cells. Data from three different experiments in sextuplicate are displayed in box-and-whisker plots in % to control and analyzed using a one-way ANOVA followed by Dunnett’s post-test. Statistical differences (*p* < 0.05) were identified by distinct letters: (a) in relation to the control; (b) in relation to H_2_O_2_, and * for differences between the experimental groups.

**Figure 5 pharmaceuticals-17-00931-f005:**
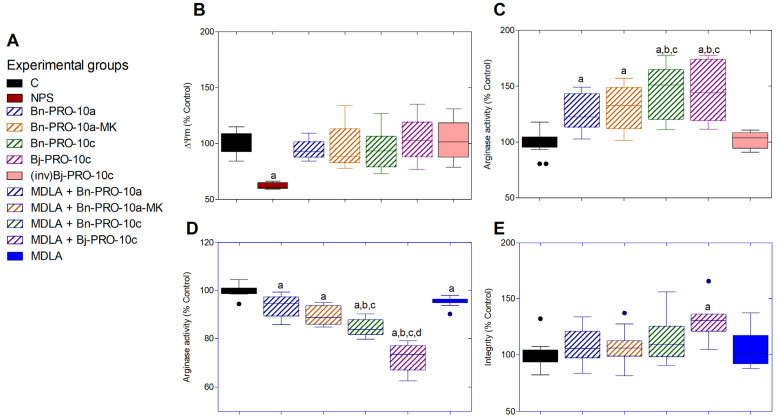
Effects of PROs on arginase activity in neuronal PC12 cells. (**A**) Schematic representation of experimental groups tested on arginase activity measured by urea concentration analyses and cell integrity using the crystal violet dye protocol. (**B**) The effects of PROs on mitochondrial membrane potential (ΔΨm). (**C**) The effects of Bn-PRO-10a, Bn-PRO-10a-MK, Bn-PRO-10c, Bj-PRO-10c, and (inv)Bj-PRO-10c on arginase indirect activity after 24 h of treatment. (**D**,**E**) The specific arginase activity and cell integrity effects shown by distinct PROs were also tested in the presence of methyl-DL-aspartic acid (MDLA), an inhibitor of argininosuccinate synthase (AsS). Data are shown as box-and-whisker plots in % to control from three independent experiments in sextuplicate and analyzed by a one-way ANOVA followed by Dunnett’s post-test. Statistical differences (*p* < 0.05) were identified by distinct letters: (a) in relation to the control; (b) in relation to Bn-PRO-10a; (c) in relation to Bn-PRO-10a-MK; and (d) Bn-PRO-10c.

**Table 1 pharmaceuticals-17-00931-t001:** Structure–activity properties of the proline-rich oligopeptides (PROs) studied. The identification of peptides was based on the number of amino acids in their primary structure, the order of their discovery, and the snake species that was found. The differential effects of PROs are presented on the MW (Da), ACE inhibition, ∆MAP, neuroprotection index, and AsS activation. PRO: proline-rich oligopeptide; MW: molecular weight; ACE: angiotensin-converting enzyme; MAP: mean arterial pressure; # peptide sequence in ClustalW format; and underline mean showing that substitutions have been observed. The symbol “+” represents the significance of the effects of PROs on AsS activation in relation to control. a: Kodama et al., 2015 [17]; b: Hayashi and Camargo, 2005 [13].

Peptide Identification	Sequence ^#^	Snake Species	MW (Da)	ACE (Ki; nM)	∆MAP mmHg	Neuroprotection Index	AsS Activation
Bn-PRO-10a	<E N WP R P K IPP	*Bitis nasicornis*	1216.7 ^a^	0.48 ^a^	−18.8 ± 0.3 ^a^	0.65	+
Bn-PRO-10a-MK	<E N WP R P K IPPM K	*Bitis nasicornis*	1476.8 ^a^	>100 ^a^	−13.9 ± 0.4 ^a^	0.90	++
Bn-PRO-10c	<E N WP R P K VPP	*Bitis nasicornis*	1202.6 ^a^	0.25 ^a^	−18.7 ± 1.2 ^a^	0.84	+++
Bj-PRO-10c	<E N WP H P Q IPP	*Bothrops jararaca*	1196.3 ^b^	0.20 ^a,b^	−18.2 ± 3.1 ^a^	0.70	++++

## Data Availability

All data generated or analyzed during this study are included in this published article.

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
