# Peer review of "Small Structural Differences in Proline-Rich Decapeptides Have Specific Effects on Oxidative Stress-Induced Neurotoxicity and L-Arginine Generation by Arginosuccinate Synthase"

_pharmaceuticals, 2024, doi:10.3390/ph17070931_

Round 1

Reviewer 1 Report

Comments and Suggestions for Authors

1. Introduction

Lines 42 - 57 - I suggest to pay attention here, highlight for the reader the importance of the cited cell models under the influence of the mentioned LMMFs, that is, for what reason did the cited research teams use them? not just list the cell lines as such. This is important because, as below, in lines 72 - 74, appears this kind of continuation of thought.

Line 107 - please consider a brief explanation of why the authors propose to use the neuronal PC12 cell line for the study - that is, why exactly this line, which is to present of what kind of model - i.e. only simulating of neuron action, function as it is? 

4. Materials and Methods

4.2. Cell Line, Culture and Maintenance

Line 354 - please consider adding information after which passage the PC12 cells were used for downstream analyses.

4.5. Neuroprotection Assay in PC12 Cells against Oxidative Stress

Line 383 - 384 - based on what data was the concentration of PROs (6.25 μmolL-1) taken for analysis that, as, what was mentioned, rescued the H2O2-induced neurotoxicity in an applied experimental model? Perhaps you might want to add here a short info based on obtained results (description - line: 143). Please give this some thought. 

4.5.3. Arginase Indirect Activity

I am not sure I understood the initial description of the apllied assay with reference to Figure 4E in the Results section. What does urea mean? please clarify the statement so that it is clear to the reader whether the assay described is a kind of model supposedly showing the transformation to a key metabolite into the urea, or whether it was added separately to a cell line culture - I suggest you clearly specify here, also here what cell line was used?.  

2. Results

What about the other two peptides? The’re not discussed in the appropriate sections of the Results chapter (primarily to the results presented in Fig. 4).

The discussion seems to be concise enough, intriguing. 

What is lacking a bit, in my opinion, is a clear indication, summary of which of the peptides studied showed, in general, the strongest preventive, neuroprotective activity with respect to the perturbing agent (H2O2). 

Please consider whether it would be appropriate to highlight in the discussion the need for further peptide sequence-dependent studies that would jendidly indicate the key protective role of the activity of the relevant amino acid motif. 

In my opinion, the statistically significant properties of the studied peptides have a smaller impact in relation to negative controls (and these (that kind of results) were not uncommon) and a more protective nature towards the disturbing, toxic factor, mainly H2O2 - and this character is particularly worth emphasizing more. 

Particularly interesting were the results described in Figure 5, especially in tabs D, E - in point D especially there are significant differences not only against toxic H2O2 but also against analyzed PROs

These are loose comments, suggestions for the authors to consider.

Author Response

Comment 1: Lines 42 - 57 - I suggest to pay attention here, highlight for the reader the importance of the cited cell models under the influence of the mentioned LMMFs, that is, for what reason did the cited research teams use them? not just list the cell lines as such. This is important because, as below, in lines 72 - 74, appears this kind of continuation of thought.

Response 1: Thank you for your interesting observation. Indeed, we have demonstrated that the LMMF (<10 kDa) obtained from “Bothrops jararaca” (Bj-LMMF) snake venom showed neuroprotective efficacy against H2O2-induced oxidative stress in neuronal PC12 cells and primary cultured hippocampus cells. However, no effects were observed in neuronal SH-SY5Y cells or C6 astrocyte-like cells. These results suggest that the cell lines types used to investigate snake venom compounds-mediated neuroprotection show important physiological differences with the cell type from which they were derived, and can influence the neuroprotective effects distinct against oxidative stress. In agreement with you, we have added new information to the revised manuscript.

Comment 2: Line 107 - please consider a brief explanation of why the authors propose to use the neuronal PC12 cell line for the study - that is, why exactly this line, which is to present of what kind of model - i.e. only simulating of neuron action, function as it is? 

Response 2: PC12 cells are derived from a pheochromocytoma of the adrenal medulla, which represents typical neuronal cells functional of the peripheral nervous system with dopaminergic properties. The use these type cells has been reported in our previous studies, according to present in the introduction section, and to be an appropriate model to explore the neuroprotective effects of snake venom compound such as specific peptide or peptide fraction of venom. The revised manuscript included the additional detail.

Comment 3: Line 354 - please consider adding information after which passage the PC12 cells were used for downstream analyses.

Response 3: We cultured the cells and used them for the experiments up to a maximum passage number of 30. The revised manuscript included the additional detail.

Comment 4: Line 383 - 384 - based on what data was the concentration of PROs (6.25 μmolL-1) taken for analysis that, as, what was mentioned, rescued the H2O2-induced neurotoxicity in an applied experimental model? Perhaps you might want to add here a short info based on obtained results (description - line: 143). Please give this some thought. 

Response 4: Your statement is accurate. Additional information on the cell integrity tests was provided to elucidate the application of a concentration of 6.25 μmol.L-1 for studying the protective effects of PRO on metabolic activity, ROS generation, and arginase activity.. Please verify the revised manuscript.

Comment 5: I am not sure I understood the initial description of the apllied assay with reference to Figure 4E in the Results section. What does urea mean? please clarify the statement so that it is clear to the reader whether the assay described is a kind of model supposedly showing the transformation to a key metabolite into the urea, or whether it was added separately to a cell line culture - I suggest you clearly specify here, also here what cell line was used?.

Response 5: Thank you for your observation. Indeed, the reference to Figure 4E is unclear. We revised the information based on the details presented in Section 4.5.3. Please check in on the revised manuscript.

Comment 6: What about the other two peptides? The’re not discussed in the appropriate sections of the Results chapter (primarily to the results presented in Fig. 4).

Response 6: The peptides Bn-PRO-10a and Bj-PRO-10c did show neuroprotective effects against H2O2-induced oxidative stress in PC12 cells, as demonstrated in figures 3A and D, respectively. To find out more about their protective effects, only Bn-PRO-10a-MK and Bn-PRO-10c with neuroprotective activity at 6.25 μmol/L were tested in studies of metabolic activity, cell integrity, ROS generation, and arginase activity.

Comment 7: The discussion seems to be concise enough, intriguing. 

Response 7: Thank you so much for appreciated our study.

Comment 8: What is lacking a bit, in my opinion, is a clear indication, summary of which of the peptides studied showed, in general, the strongest preventive, neuroprotective activity with respect to the perturbing agent (H2O2).

Response 8: In our investigation, we found that only Bn-PRO-10a-MK and Bn-PRO-10c showed a dose-dependent ability to restore cell viability against oxidative stress damage, whereas Bn-PRO-10a and Bj-PRO-10c did not exhibit this capability despite their significant similarity to the tested PROs. We believe that independent mechanisms could explain the neuroprotective effects of both peptides, and we prefer not to explore how they show the most effect. Our results open new perspectives to investigate in more detail how these peptides work against oxidative stress, using representative inhibitors of important neuroprotection pathways to predict possible mechanisms of action. Curiously, Bn-PRO-10c (<ENWPRPKVPP) and Bj-PRO-10c (<ENWPHPQIPP) show different biological effects (Table 1), suggesting that, aside from both peptides increasing L-arginine production, the particular Bn-PRO-10c-mediated neuroprotection appears to be independent of the L-arginine metabolism pathway reported by Bj-PRO-10c. Additional information on the explanation was incorporated into the discussion section.

Comment 9: Please consider whether it would be appropriate to highlight in the discussion the need for further peptide sequence-dependent studies that would jendidly indicate the key protective role of the activity of the relevant amino acid motif.

Response 9: Indeed, further research is necessary to gain a more comprehensive understanding of the correlation between the structure and functions of PROs, as well as their ability to protect against oxidative stress. We incorporated additional information into the discussion section.

Comment 10: In my opinion, the statistically significant properties of the studied peptides have a smaller impact in relation to negative controls (and these (that kind of results) were not uncommon) and a more protective nature towards the disturbing, toxic factor, mainly H2O2 - and this character is particularly worth emphasizing more.

Response 10: Indeed, Bn-PRO-10a-MK and Bn-PRO-10c-mediated neuroprotection appear to not restore basal levels seen in the control group (Figure 3), especially when Bn-PRO-10c is used. However, it is crucial to note that our study employed a one-way analysis of variance (ANOVA) for between-groups comparisons, a Dunnett's post-hoc test for multiple comparisons, and a representative number of assays and repetitions, each consisting of three independent experiments in sextuplicate. Nevertheless, our findings provide potential to expand the inquiry by using other experimental conditions.

Comment 11: Particularly interesting were the results described in Figure 5, especially in tabs D, E - in point D especially there are significant differences not only against toxic H2O2 but also against analyzed PROs

Response 11: Yes, it was interesting how small differences in the structure of PROs can affect availability of L-arginine. Our group has initiated additional research based on these findings in order to verify the presence of these changes in a more complex experimental model using zebrafish up to 5 days after fertilization. This model has been used in our laboratory to study Parkinson's disease.

Comment 12: These are loose comments, suggestions for the authors to consider.

Response 12: We appreciate your valuable comments and suggestions about our study, which were important to preparing this revised manuscript.

Reviewer 2 Report

Comments and Suggestions for Authors

The authors describe how small structural variations may contribute to protection of oxidatively induced neurotoxicity. This study is thorough and the data supports their conclusions. I recommend publish after some minor revisions.

Comment:

I am curious about the structural similarities between the peptides and how they interact with the AsS enzyme. Did you try any ligand-protein docking to interrogate the interactions between the enzyme and the peptide and how these lead to increased or decrease affinity of the enzyme for the peptide?

Author Response

We appreciate your effort spent reviewing our study, as well as your comments and words of support.

Comment 1: I am curious about the structural similarities between the peptides and how they interact with the AsS enzyme. Did you try any ligand-protein docking to interrogate the interactions between the enzyme and the peptide and how these lead to increased or decrease affinity of the enzyme for the peptide?

Response 1: It is interesting to understand that small modifications in the peptides studied in this study could either improve or reduce their impact on AsS function. The ligand-protein docking investigations have not yet started; however, they are currently being scheduled for implementation in the near future.